# Function and Regulation of Chloroplast Peroxiredoxin IIE

**DOI:** 10.3390/antiox10020152

**Published:** 2021-01-21

**Authors:** Anna Dreyer, Patrick Treffon, Daniel Basiry, Anna Maria Jozefowicz, Andrea Matros, Hans-Peter Mock, Karl-Josef Dietz

**Affiliations:** 1Department of Biochemistry and Physiology of Plants, Faculty of Biology, University of Bielefeld, 33615 Bielefeld, Germany; anna.dreyer@uni-bielefeld.de (A.D.); ptreffon@umass.edu (P.T.); danielbasiry@gmx.de (D.B.); 2Applied Biochemistry Group, Leibniz Institute for Plant Genetics and Crop Plant Research (IPK), 06466 Gatersleben, Germany; jozefowicz@ipk-gatersleben.de (A.M.J.); andrea.matros@adelaide.edu.au (A.M.); mock@ipk-gatersleben.de (H.-P.M.)

**Keywords:** peroxiredoxin, AT3G52960, glutathione, S-glutathionylation, glutaredoxin, 14-3-3 protein, phosphorylation, posttranslational modification, redox regulatory network

## Abstract

Peroxiredoxins (PRX) are thiol peroxidases that are highly conserved throughout all biological kingdoms. Increasing evidence suggests that their high reactivity toward peroxides has a function not only in antioxidant defense but in particular in redox regulation of the cell. Peroxiredoxin IIE (PRX-IIE) is one of three PRX types found in plastids and has previously been linked to pathogen defense and protection from protein nitration. However, its posttranslational regulation and its function in the chloroplast protein network remained to be explored. Using recombinant protein, it was shown that the peroxidatic Cys121 is subjected to multiple posttranslational modifications, namely disulfide formation, S-nitrosation, S-glutathionylation, and hyperoxidation. Slightly oxidized glutathione fostered S-glutathionylation and inhibited activity in vitro. Immobilized recombinant PRX-IIE allowed trapping and subsequent identification of interaction partners by mass spectrometry. Interaction with the 14-3-3 υ protein was confirmed in vitro and was shown to be stimulated under oxidizing conditions. Interactions did not depend on phosphorylation as revealed by testing phospho-mimicry variants of PRX-IIE. Based on these data it is proposed that 14-3-3υ guides PRX‑IIE to certain target proteins, possibly for redox regulation. These findings together with the other identified potential interaction partners of type II PRXs localized to plastids, mitochondria, and cytosol provide a new perspective on the redox regulatory network of the cell.

## 1. Introduction

Chloroplasts of cormophytes contain three types of peroxiredoxins (PRXs), namely classical 2-cysteine peroxiredoxin (2-CysPRX), a bacteroferritin-comigratory protein homolog PRX-Q and a type II peroxiredoxin named PRX-IIE [1,2]. The basic PRX complement of plastids is subjected to variation by the presence of isoforms, e.g., 2-CysPRX-A and -B in *Arabidopsis thaliana* (At3g11630, At5g06290), and PRX-IIE-1 and PRX-IIE-2 in *Oryza sativa* (Os06g42000, Os02g09940) [3].

PRXs are thiol peroxidases. They possess a peroxidatic cysteinyl thiol (Cys_P_) with a very low pK value and thus expose the deprotonated thiolate anion in a conserved catalytic environment. Because of this particular feature, PRXs function as highly affine and efficient thiol peroxidases [4]. The catalytic activity of 2-CysPRX, PRX-Q, and type II PRX relies on a conserved second cysteine, which acts as resolving thiol (Cys_R_). Upon reaction with the peroxide substrate, the Cys_P_-thiol forms a sulfenic acid derivative which immediately is attacked by the Cys_R_. A disulfide bond is formed between Cys_P_ and Cys_R_. Prior to the next catalytic cycle, the disulfide bond needs to be reduced to dithiols by electron donors like thioredoxin (TRX) or glutathione/glutaredoxin (GRX). The different PRX forms show variation in their primary sequence, as well as the presence and position of Cys_R_ and the regeneration mechanism [2].

The best understood plant PRX is the 2-CysPRX, which has been scrutinized with respect to its catalytic properties, redox-dependent conformational dynamics, regeneration by electron transmitters such as TRX and GRX/GSH, and its role in the redox regulatory network of the chloroplast [5,6,7,8]. In contrast, PRX-Q and PRX-IIE have been studied for their peroxidatic property, regeneration mechanism, and their function in mutant plants with decreased protein amounts [9,10].

PRX-IIE belongs to the group of highly conserved atypical 2-cysteine peroxiredoxins, initially discovered in the phloem of poplar [11]. PRX-II type PRX are found in, e.g., some but not all photosynthetic cyanobacteria [12], animals/humans (PRDX5) [13], and lower and higher plants [14].

Previous studies showed PRX-IIE to be localized in the soluble fraction of plastids. Its transcript amount in *A. thaliana* leaves slightly increased in response to high light, decreased upon ascorbate and NaCl treatment, and remained unchanged with leaf age [1,15]. The redox titration of the dithiol-disulfide transition gave a midpoint redox potential of −288 mV which is 19 and 34 mV less negative than 2-CysPRX-A and -B [1], respectively.

Re-reduction of the oxidized PRX-IIE from poplar was most efficient with GRX/glutathione system and low with glutathione and TRX [14]. The catalytic efficiency was 10^5^ M^−1^s^−1^ with tertiary butylhydroperoxide (t-BOOH), fourfold less with H_2_O_2_, and activity was absent with cumene hydroperoxide (CuOOH) [14]. Among the three GRXs (GRX-S12, -S14, and -S16) that are targeted to the plastid in poplar, only GRX-S12 efficiently regenerated oxidized PRX-IIE [14]. The authors suggested a catalytic mechanism where the sulfenic acid derivative, formed by reaction with the peroxide substrate, reacts with reduced glutathione to form an S-glutathionylated intermediate. GRX reduces the Cys_P_-SG and becomes itself S-glutathionylated. Another glutathione molecule then regenerates GRX-SG, leading to the formation of oxidized glutathione (GSSG) [14,16].

This study aimed for the biochemical characterization of PRX-IIE, with a focus on redox-dependent posttranslational modifications of its critical cysteine residues and their impact on protein function. In the light of the presumed function in redox signaling, it appeared timely to address the interactome of PRX-IIE by affinity chromatography and mass-spectrometric identification. Among other proteins, six members of the phosphoprotein-binding 14-3-3 family were identified. Their function and activity are often connected to cell signaling function [17]. Therefore, one of them, namely the 14-3-3 υ was used to further validate the interaction. The presence of 14-3-3 υ slightly affects the peroxidase activity. These novel findings point out an additional role of PRX-IIE in redox signaling apart from its peroxidase activity.

## 2. Materials and Methods

### 2.1. Plant Material and Growth Conditions

*A. thaliana* Col-0 plants were grown on soil (SM Max Planck Köln, project 187509, Stender, Germany) in a greenhouse with 14 h day and 10 h night at 25 °C.

### 2.2. Cloning

PRX-IIE (AT3G52960), Glutaredoxin-S12 (GRX-S12; AT2G20270), and sulfiredoxin (SRX; AT1G31170) were cloned into pET28a (Novagen, Darmstadt, Germany). Forward and reverse primer were designed with *NdeI* and *BamHI* restriction sites, respectively (Appendix A). The variants C121S, C146S, C121S/C146S, S82D, T108E, and T223E of PRX-IIE were generated by site-directed mutagenesis with specific primers (Appendix A). The correctness of all constructs was confirmed by DNA sequencing.

### 2.3. Heterologous Expression and Purification

*E. coli* BL21 (DE3)pLysS cells (Invitrogen, Carlsbad, CA, USA) were transformed with plasmid DNA. Overnight cultures were used to inoculate 2 L lysogeny broth medium supplemented with 50 µg/mL kanamycin and 20 µg/mL chloramphenicol. Protein expression was induced in the exponential phase by the addition of isopropyl-β-D-thiogalactopyranoside to a final concentration of 0.4 mM. Induced cells were grown at 37 °C and 140 rpm for 4 h. Cells were harvested for 15 min at 5000× *g* and resuspended in lysis buffer (50 mM NaH_2_PO_4_, 300 mM NaCl and 10 mM imidazole, pH 8.0). Cells were disrupted using lysozyme digestion followed by sonication (HF-Generator GM2070 in combination with an ultrasonic converter UW2070, standard horn SH 70G, and microtip MS73, Bandelin, Berlin, Germany). The soluble and insoluble fractions were separated by centrifugation at 20,000× *g* for 45 min. His-tagged proteins were incubated with nickel-nitrilotriacetic acid (Ni-NTA) sepharose (Qiagen, Hilden, Germany) at 4 °C with slight shaking for 1 h. Washing was performed with wash buffer I (50 mM NaH_2_PO_4_, 300 mM NaCl, and 20 mM imidazole, pH 8.0) and subsequently with wash buffer 2 (50 mM NaH_2_PO_4_, 300 mM NaCl, 20 (*v*/*v*) glycerol, and 50 mM imidazole, pH 8.0) until the OD_280_ reached zero. Elution was achieved with elution buffer (50 mM NaH_2_PO_4_, 300 mM NaCl, and 250 mM imidazole, pH 8.0). Protein containing fractions were pooled and dialyzed against 40 mM K-Pi, pH 7.2. After dialysis, protein concentrations were determined using a molar extinction coefficient of 8480 M^−1^ cm^−1^ for PRX-IIE and its variants and 9970 M^−1^·cm^−1^ and 4470 M^−1^·cm^−1^ for GRX-S12 and SRX, respectively. GRX-C5 (At4g28730) was purified following established procedures [18,19].

### 2.4. Xylenol Orange Assay

The ferrous-dependent xylenol orange assay (FOX) was used to analyze the activity of PRX-IIE and its phosphomimic variants with DTT as electron donor. The reaction mixture contained 2 µM PRX-IIE and 4 mM dithiothreitol (DTT) in 40 mM K-Pi, pH 7.2. The measurement was started by the addition of peroxides (400 µM H_2_O_2_, 200 µM t-BOOH, or 200 µM CuOOH) in a time course of 90 s at 15 s intervals. The remaining peroxides were detected by ferrous-dependent oxidation of xylenol orange as reported previously [20].

### 2.5. NADPH-Dependent Peroxidase Activity Measurement

The reduction of peroxides by PRX-IIE was monitored with the GRX system as reductant. The activity was measured using a Cary 300 Bio UV/VIS spectrometer (Varian, Middelburg, The Netherlands) following NADPH oxidation at 340 nm. The assay was performed at 25 °C in quartz cuvettes with 2 µM PRX-IIE, 0.5 units glutathione reductase (GR), 200 µM NADPH, 1 mM EDTA, 1 mM GSH, varying amounts of GRX-S12 and peroxides (H_2_O_2_, t-BOOH, CuOOH) in 40 mM K-Pi, pH 7.2.

### 2.6. Hyperoxidation of PRX-IIE

Hyperoxidation of PRX-IIE was assayed as described above using the FOX assay with 400 µM H_2_O_2_ as substrate and increasing CuOOH concentrations. Furthermore, the oxidation state was investigated by electrospray ionization coupled with mass spectrometry (ESI-MS; Esquire 3000, Bruker Daltonics, Bremen, Germany). 10–20 µM of prereduced protein in 100 mM Tris-HCl, pH 8.0, was incubated with 5 mM DTT and different CuOOH concentrations or 0.5 mM DTT and increasing H_2_O_2_ concentrations for 1 h at room temperature (RT). Excess low molecular weight reagents were removed by acetone precipitation and proteins were resuspended in H_2_O. Dilutions were prepared in 30% EtOH, 0.1% formic acid (FA) and the mixture was introduced into the ESI-MS. Instrumental settings: Capillary voltage = 4.000 V. Nebulizer gas pressure = 15 psi. Drying gas flow = 4.0 L/min. Drying gas temperature = 300 °C. Mass-to-charge (m/z) values: 650-1200. Mass spectra were deconvoluted using the software provided by the manufacturer (DataAnalysis, Bruker Daltonics, Bremen, Germany).

### 2.7. S-Glutathionylation

10–30 µM PRX-IIE in 100 mM Tris-HCl, pH 8, was reduced for 30 min at room temperature (RT) with 4 mM DTT. Desalting was achieved by passing the solution through PD10 columns. S-glutathionylation was carried out by disulfide exchange with oxidized glutathione (GSSG) for 1 h at RT. Excess GSSG was removed via acetone precipitation. Afterward, S-glutathionylation was detected by Western blot using a monoclonal anti-GSH antibody (Thermo Scientific, Schwerte, Germany). In addition, molecular masses of modified and unmodified proteins were assessed by ESI-MS as mentioned before. For deglutathionylation, 10 µM glutathionylated PRX-IIE was incubated with 10 µM of GRX-S12, GRX-C5, or SRX and 0.5 mM GSH at 25 °C. The decrease of glutathionylated PRX-IIE was determined using Western Blot with anti-GSH antibodies. The spot intensities were analyzed using ImageJ.

### 2.8. 2-Dimensional SDS-PAGE

6-week-old *A. thaliana* Col_0 plants were sprayed with 300 µM methylviologen (MV) and 0.1% (*v*/*v*) Tween-20 as control, respectively. After 3 h the plants were harvested and immediately frozen in liquid nitrogen and ground to a fine powder. Proteins were isolated and used for the separation in the first dimension with Immobiline Dry Stripes (pH range 3–10 NL, 18 cm, GE Healthcare, Uppsala, Sweden) [21]. 250 µg protein were dissolved in 340 µL rehydration buffer (0.01% ampholyte; 0002% (*w/v*) bromophenol blue) and applied to the Immobiline Dry Stripe. The rehydration and focusing consisted of the following steps: 1 h 0 V, 12 h 30 V, 2 h 60 V, 1 h 500 V, 1 h 1000 V, and finally 8000 V for as long as needed to reach 42,000 Vh. Separation in the second dimension was done with 12% non-reducing SDS-PAGE. Afterward, the gel was blotted to nitrocellulose membrane and subjected to Western blotting with PRX-IIE antibody and peroxidase-labeled secondary antibodies. Detection was done with ECL Substrate (GE Healthcare, Chicago, IL, USA) and X-ray films.

### 2.9. Subcellular Localisation of PRX-IIE

The open reading frame of PRX-IIE from *A. thaliana* was cloned into the 35S-EYFP-NosT vector using specific primers (Appendix A) for in vivo subcellular localization of PRX-IIE [22]. The resulting construct consisted of the PRX-IIE preprotein fused to EYFP as a reporter under the control of the CaMV35S-promoter. Transient expression in mesophyll protoplasts and confocal laser scanning microscopy were performed as described before [23,24].

### 2.10. Affinitiy Chromatography and Mass Spectrometry

Reduced His-tagged PRX-IIE (3 mg) or PRX-IIE C146S (3 mg) were bound to 1 mL Ni-NTA resin (Qiagen, Hilden, Germany) and used as an affinity matrix. Ni-NTA matrix without PRX-IIE served as control. Leaves of about 5-week-old plants were homogenized in 50 mM Tris-HCl, pH 8.0, 1 mM PMSF, and afterward, filtrated through Miracloth. Clear protein extract was obtained via centrifugation (30 min at 20,000 rpm and 4 °C). The supernatant (about 40 mg protein) was applied to the matrix and incubated at RT with gentle agitation for 1.5 h. Non-bound material was removed by washing the column with 20 mL of 50 mM Tris-HCl, pH 8.0, and 20 mL of 50 mM Tris-HCl, pH 8.0, 200 mM NaCl. The first elution step was achieved with 1 mL of 50 mM Tris-HCl, pH 8.0, 200 mM NaCl, 50 mM DTT, and incubation for 15 min at RT. The eluted fraction was collected and stored. Afterward, the columns were washed with 10 mL 50 mM Tris-HCl, pH 8.0, 200 mM NaCl. The second elution step was achieved by using a high concentration of imidazole, therefore 1 mL 50 mM Tris-HCl, pH 8.0, 200 mM NaCl, 50 mM DTT, and 500 mM imidazole were applied to the column and incubated at RT for 15 min. The second elution step was also collected and stored. The samples of the first and second elution were trypsinated after chloroform/methanol precipitation [25] and dissolved in 0.1% formic acid, 1% acetonitrile. Peptides were separated by reverse-phase nano-LC and analyzed by electrospray ionization-mass spectrometry (ESI-QTOF-MS) as described [26]. Data were searched against the entries of UP000006548 3702 ARATH *A. thaliana* of the UniProt database using ProteinLynx Global Server 3.0.2. Proteins, which were found in two out of three biological experiments with at least two peptides were accepted for further analysis. In addition, proteins, which were identified in the control sample (nonspecific binding), were removed from protein lists. The LC-MS data are deposited using the e!DAL system of IPK Gatersleben [27] and available at http://dx.doi.org/10.5447/ipk/2021/0 [28]. 

### 2.11. Far Western Blot

Dilution series of recombinant 14-3-3 υ protein were spotted on nitrocellulose membrane, together with a dilution series of PRX-IIE as a calibration curve. The membrane was blocked with Tris-buffered saline (TBS), pH 7.5, containing 1% (*w/v*) milk powder. The membrane was incubated with PRX-IIE or its phospho-mimicry variants in TBS with either 1 mM DTT or 100 µM H_2_O_2_ overnight at 4 °C. After three times washing with TBS for 5 min, proteins were detected using a specific anti-PRX-IIE antibody, a peroxidase-labeled secondary antibody against rabbit, ECL^®^ substrate (GE Healthcare, Chicago, IL, USA), and X-ray films. Intensity quantification of the spots after documentation was done using ImageJ. The relative amounts of bound PRX-IIE were determined in the linear range of the blots.

### 2.12. Structural Modeling

PRX-IIE structure was obtained from SWISS-MODEL [29] based on the structure of *Populus tremula* PRX D type II (pdb: 1tp9A). Further analysis was done with PyMOL version 2.4.0 [30].

### 2.13. Statistical Analysis

Statistics were calculated either using F-Test followed by students T-test or using one-way ANOVA together with post hoc Tukey honest significance differences (HSD) test. Statistical results obtained with the students T-test are indicated in the figures using one to three asterisks representing the different p-values (*: *p* < 0.05; **: *p* < 0.01; ***: *p* < 0.001). Statistical results using the one-way ANOVA together with post hoc Tukey HSD are represented using different letters using *p* ≤ 0.05.

## 3. Results

### 3.1. PRX-IIE Is Localized to the Chloroplast Stroma

The PRX family in *A. thaliana* consists of 10 ORFs, of which 9 members are described to be expressed [31]. Besides the mitochondrial PRX-IIF, PRX-IIE is the only type II PRX in *A. thaliana* which displays a putative transit peptide covering for the first 70 amino acids (Figure 1A). The remaining amino acids form the stable TRX-like structure with seven β-sheets and five α-helices (Figure 1B). To confirm the subcellular localization of PRX-IIE, a plasmid encoding the PRX-IIE-EYFP fusion protein was transfected into mesophyll protoplasts. Confocal laser scanning microscopy of the transfected protoplasts revealed the plastidial localization, as indicated in the overlay of the EYFP signal with the chlorophyll autofluorescence (Figure 1C).

### 3.2. PRXIIE Detoxifies H_2_O_2_ Using the GRX System for Regeneration

Peroxiredoxins reduce a broad range of peroxides and their activities rely on the conserved cysteine residues. The reduction of H_2_O_2_ by PRX-IIE and its cysteine variants was determined using the FOX assay. The removal of either the Cys_p_ as well as the Cys_R_ residue had a negative impact on peroxidase activity, constraining that both thiol groups are necessary for reactive oxygen species (ROS) scavenging (Figure 2A). Substrate specificity with H_2_O_2_, t-BOOH, and CuOOH of wild-type (WT) protein is depicted in Figure 2B. PRX-IIE showed the highest rate of activity with H_2_O_2_ as substrate and DTT as reductant. Reduction rates of t-BOOH and CuOOH relative to H_2_O_2_ were about 20% and 0.5%, respectively. This indicates that H_2_O_2_ is the preferred substrate for PRX-IIE. Furthermore, peroxide reduction by PRX-IIE in the presence of chloroplastic glutaredoxin-S12 (GRX-S12) as reductant was determined (Figure 2C) and confirmed H_2_O_2_ as the preferred substrate, but unlike with DTT, lower activities were recorded.

### 3.3. CuOOH-and H_2_O_2_-Dependent Hyperoxidation

CuOOH is a strong oxidizing agent and based on activity measurements (Figure 2B) it was assumed that PRX-IIE is hyperoxidized by CuOOH. This hypothesis was tested in vitro using the FOX assay at increasing CuOOH concentrations (Figure 3A). Significant inhibition of H_2_O_2_ detoxification could be detected in the presence of 12.5 µM CuOOH and the peroxidase activity was undetectable at high CuOOH concentration. To further address the possibility for hyperoxidation of PRX-IIE by CuOOH, ESI-MS analyses were carried out (Figure 3B). In contrast to reduced PRX-IIE with a molecular mass of 19,438.0 Da, the sample treated with 0.5 mM CuOOH showed a mass increase of ~32 Da corresponding to the formation of the sulfinic acid derivative (-SO_2_H). Higher CuOOH concentrations lead to further oxidation to the sulfonic acid (-SO_3_H).

ESI-MS analysis of cysteine variants of PRX-IIE was performed to elucidate which of the two cysteines is prone to redox modifications (Table 1). Deconvoluted data revealed that hyperoxidation occurs in the C146S variant, lacking the Cys_R_, while the PRX-IIE protein variant lacking Cys_p_ (C121S) showed only slight oxidation at high CuOOH concentrations. This assay relied on a single turnover of peroxide reduction and, therefore, used relatively high CuOOH concentrations in a first set.

Activity measurements for PRX-IIE revealed that H_2_O_2_ is the preferred substrate (Figure 2B), but H_2_O_2_ is also known to catalyze hyperoxidation of proteins [19]. To test this for PRX-IIE, peroxide-mediated hyperoxidation was analyzed by ESI-MS (Figure 4). The extent of sulfinic acid formation was calculated from the ratio of the peak intensities for the reduced (-SH, 19,438.0 Da) and hyperoxidized (-SO_2_H, 19,470.0 Da) protein in the deconvoluted ESI-MS spectra. Incubation of PRX‑IIE with 25 µM H_2_O_2_ resulted in an hyperoxidation rate of 46%, and higher peroxide concentrations lead to further hyperoxidation of the protein. Masses that correspond to the sulfonic acid (-SO_3_H) could not be detected, suggesting that H_2_O_2_-mediated hyperoxidation is limited to the formation of the sulfinic acid derivative at Cys_P_.

### 3.4. S-Glutathionylation of PRX-IIE Occurs at Cys_p_

The results shown above revealed a lower peroxidase activity for PRX-IIE with GRXs as an electron donor in comparison to DTT as reductant (Figure 2C). The antioxidant glutathione is one component of the assay. In addition, H_2_O_2_-dependent hyperoxidation of PRX-IIE could be observed (Figure 4). For *A. thaliana* and *T. brucei* reversible S-glutathionylation, the addition of one glutathione molecule to specific cysteine residues has been shown to prevent 2-Cys PRX hyperoxidation and thereby regulates its function [32,33]. To analyze the possibility for this redox-related posttranslational modification, reduced PRX-IIE was incubated with either 0.5 mM DTT or 10 mM oxidized glutathione (GSSG) overnight at 4 °C. Following acetone precipitation samples were subjected to ESI-MS and masses of the intact protein were obtained (Figure 5)

Deconvoluted data revealed the addition of one glutathione molecule that increased the mass of the protein by 306 Da (Figure 6A). The PTM of PRX-IIE was further proven with a specific anti-GSH antibody (Figure 6B). In addition, S-glutathionylation of PRX-IIE at low physiological GSSG concentrations could be observed (Figure 6A). The time-dependent S-glutathionylation in vitro was detected after incubation with GSSG for different time points (Figure 6C). Thiol modification was observed already after 10 min and reached a maximum of 60 min. The results demonstrate the fast S-glutathionylation of PRX-IIE at physiologically relevant concentrations in vitro. Not only incubation of PRX-IIE with GSSG resulted in S-glutathionylation, but also incubation of pre-reduced PRX-IIE with 1 or 5 mM S-nitrosoglutathione. Besides this, the formation of S-nitrosation (19.469 kDA, -SNO) and S-nitrosoglutathionylation (19.772 kDa, -SSGNO) was observed, using ESI-MS (Appendix A).

To test for the particular residue that is prone to S-glutathionylation, Cys to Ser variants were analyzed using Western Blot with the specific anti-GSH antibody. Results showed S-glutathionylation of WT PRX-IIE and the C146S variant, but not for the C121S variant (Figure 7). Furthermore, mass spectrometry was done with WT and the Cys→Ser variants. 19,422 Da corresponds to the reduced variants with single mutated cysteine residues (C121S or C146S) and 19406 Da to the double variant. Only the C146S protein, lacking the Cys_R_ at position 146, was S-glutathionylated in a concentration-dependent manner using GSSG (Figure 7B).

Next, the effect of S-glutathionylation on peroxidase activity was investigated using a modified FOX assay (Figure 8A). Peroxidase activity in the presence of GRX-S12, GSH, and H_2_O_2_ as substrate decreased in the presence of GSSG amounts as low as < 2.5% GSSG of total glutathione. The extent of glutathionylation was tested by ESI-MS at different GSH/GSSG ratios (Figure 8B). Deconvoluted spectra showed the presence of PRX-IIE-SG already at 1.25% GSSG, which is in line with the decreased peroxidase activity. Protein-SG forms through several mechanisms in vitro, however the precise reaction mechanism in vivo remains unclear [34]. In contrast, deglutathionylation reaction is reported to be catalyzed by GRXs and SRXs [35,36]. To monitor the deglutathionylation of PRX-IIE, PRX-IIE-SG was incubated with GSH in the presence or absence of GRX-S12, GRX-C5, and SRX (Figure 8C). Western blot analysis with specific GSH antibody and subsequent analyses of spot intensities using ImageJ revealed a decrease in signal intensity for PRX-IIE-SG already after 10 min of reaction time with GSH alone (Figure 8C and Appendix A). The addition of SRX or GRX did not increase the deglutathionylation reaction, leading to the conclusion that the glutathione pool itself is capable of modulating the redox state of PRX-IIE.

Protein S-glutathionylation ensures the protection of critical protein thiols against irreversible hyperoxidation in vivo and is therefore considered a biomarker for oxidative stress [37,38]. To test for this redox-dependent posttranslational modification of PRX-IIE in vivo, *A. thaliana* plants were sprayed with a single high dose of 300 µM methylviologen, harvested after 3 h, and analyzed using non-reducing two-dimensional gel electrophoresis following Western blot with the specific anti-PRX-IIE antibody. In contrast to the mock-treated samples, which showed two protein spots representing reduced (-SH) and presumably oxidized protein species, plants stressed with MV exhibited three distinct protein spots, two of them differed in acidity and molecular mass (Figure 9). Mature PRX-IIE-SH displays a molecular mass of 17,260 Da with a theoretical pI value of 5.02, whereas for PRX-IIE-SG a mass shift to 17,566 Da and a more acidic pI value of 4.91 are predicted. Molecular masses and pI values were obtained using the Expasy ProtParam tool [39]. Theoretical pI calculations and correlation with the pI values observed on the 2D gels together with the mass change suggest that the acidic spots of the triplet correspond to the glutathionylated protein.

### 3.5. Identification of PRX-IIE Interaction Partners

The next experiments aimed to elucidate whether PRX-IIE exclusively functions as a peroxidase or if it might be involved in cell signaling by direct protein-protein interactions. For that, an affinity chromatography approach with immobilized His_6_-tagged PRX-IIE and total *A. thaliana* leaf protein was used to identify interacting proteins. After washing, bound proteins were eluted with buffer containing DTT or imidazole and subsequently identified by mass spectrometry. In addition, a thiol-trapping experiment with the C146S variant was used to identify redox-regulated proteins that preferentially interact through Cys_p_. The reductive elution with DTT resulted in the identification of 47 proteins, 14 of which were also identified to interact with the C146S variant (Figure 10A). To address proteins that interact with PRX-IIE electrostatically, a second elution was done with imidazole. Seven proteins co-eluted with PRX-IIE, and 9 with the PRX-IIE C146S variant (Figure 10B). Out of the 54 proteins that interacted with PRX-IIE, 24 are localized within the chloroplast, and half of them are involved in metabolic processes (Figure 10C). However, in comparison to the WT protein, almost the same number of proteins were trapped by the C146S variant, but only 14 of them reside in the chloroplast (Figure 10D).

Interestingly, 4 out of the 24 identified chloroplast proteins trapped by the WT PRX-IIE protein belong to the 14-3-3 family (Table 2). In total 6 different 14-3-3 proteins were identified regardless of the bait protein (Appendix A) [40,41,42,43,44,45,46,47,48,49,50,51]. Plant 14-3-3 proteins are reported to be involved in multiple developmental and stress-related processes, such as apoptosis, leaf shape, and salt stress tolerance [52,53,54]. They normally occur as homodimers or heterodimers and can bind two different targets at the same time and, therefore, act as scaffold proteins [55]. Furthermore, they are involved in signaling processes regulated by phosphorylation [48,56,57]. Since PRX-IIE has three experimentally reported phosphorylation sites [58], an interaction between phosphorylated PRX-IIE and a 14-3-3 protein could be part of a signaling process. 14-3-3 υ was chosen as a representative to study the interaction between PRX-IIE and 14-3-3, because of the high number of identified unique peptides and the reported chloroplastic localization [48] (Appendix A). Therefore, *A. thaliana* 14-3-3 υ was expressed, purified, and used in further experiments.

To characterize the interaction between 14-3-3 υ and PRX-IIE, phospho-mimicry variants of PRX-IIE were generated and their peroxidase activity was analyzed using the FOX-Assay. Interestingly, all variants revealed lower peroxidase activity in comparison to WT PRX-IIE, whereas S82D and T108E showed a significantly lower activity (Figure 11A). Binding between PRX‑IIE or its phosphomimic variants and 14-3-3 υ under defined redox conditions was assessed in an overlay approach (Figure 11B). Binding was similar under all conditions apart from the 2.5-fold improved binding of WT PRX-IIE to 14-3-3 υ under oxidizing conditions (Figure 11B). Supplementation with 14-3-3 υ had a beneficial effect on peroxidase activity albeit a ten-fold excess of 14-3-3 υ was necessary to observe a significant increase of PRX-IIE peroxidase activity (Figure 11C).

## 4. Discussion

PRX-IIE is a thiol-dependent peroxidase that is localized to the chloroplast stroma (Figure 1C). Two cysteinyl residues are highly conserved within the type II PRX (Figure 1A) and mutation of either one or both cysteines to serine results in a loss of peroxidase activity (Figure 2A). Therefore, both Cys, namely Cys_p_121 and Cys_R_146, are essential for the PRX-IIE peroxidase activity.

The human homolog peroxiredoxin 5 (PRDX5) more readily reduces t-BOOH [59], while PRX-IIE from *A. thaliana* showed the highest activity with H_2_O_2_ as a substrate (Figure 2B,C). The differences in substrate specificity could be due to the different accessible surface areas of Cys_p_ of both enzymes. The accessible surface area of the Cys_p_ of PRDX5 is 1.305 Å^2^, whereas the accessible surface area of the PRX-IIE Cys_p_ is just 0.795 Å^2^ (Figure 1B). Therefore, PRX-IIE seems more likely to detoxify smaller peroxides in comparison to the human analog PRDX5.

### 4.1. Regeneration of Reduced PRX-IIE Limits Catalytic Turnover

The GRX-S12-coupled assay revealed the same substrate preference as the FOX assay, however, the catalytic activities were lower in comparison to the DTT-driven activity, especially in the case of H_2_O_2_ as substrate (Figure 2C). Since activity measurements with t-BOOH showed almost the same values, it seems that PRX-IIE regeneration by GRX-S12 is the rate-limiting step in the catalytic cycle of peroxide reduction, disulfide formation, and regeneration.

### 4.2. Bulky Substrates Favour Hyperoxidation and Inhibition of PRX-IIE

Activity was undetectable in both assays using CuOOH as substrate, and rather bulky substrates like CuOOH inhibited peroxidase activity, which was already reported for *poplar* PRX-IIE [14]. The inhibitory effect of CuOOH on the peroxidase activity of PRX-IIE was revealed by a decrease in H_2_O_2_-reduction activity in the presence of increasing CuOOH concentrations. Already the presence of 12.5 µM CuOOH resulted in a significant decrease in PRX-IIE peroxidase activity (Figure 3A) and the inhibition of activity correlated with hyperoxidation of PRX-IIE (Figure 3B). Since Figure 3A represents an activity assay in presence of DTT, the oxidized PRX-IIE (SO_2_H) accumulates during the catalytical activity [60]. Therefore, a lower amount of CuOOH is needed in comparison to the ESI-MS measurements (Figure 3B).

ESI-MS measurements of CuOOH-treated PRX‑IIE C121S and C146S variants showed the hyperoxidation at C121 to sulfinic (-SO_2_H) and sulfonic acid (SO_3_H) at intermediate amounts of CuOOH (Table 1). Oxidation of the resolving thiol C146 to sulfenic acid (-SOH) occurred after treatment with relatively high CuOOH concentrations only. H_2_O_2_ treatment resulted in oxidation of PRX-IIE to the sulfinic acid derivative during continuous thiol peroxidase cycling (Figure 4) similar to 2-CysPRX, where hyperoxidation occurs after about 250 peroxidase cycles (Figure 4) [61]. Oxidized sulfenylated 2-CysPRX functions in proximity-based oxidation reactions. Sobotta et al. [62] reported this type of signaling cascade, where human PRDX2 gets oxidized by ROS and afterward oxidizes STAT3. Disulfide-bonded 2-CysPRX from *A. thaliana* oxidizes chloroplast TRXs which in turn oxidize target proteins in the Calvin-Benson cycle or malate dehydrogenase [8]. This type of regulation was termed TRX oxidase function of 2-CysPRX and participates in the adjustment of enzyme activity to decreased light intensity. PRX-IIE is, therefore, suggested to be part of a ROS-induced signaling cascade, whereas PRX-IIE may oxidize a nearby protein.

Hyperoxidized PRX may also function in cell signaling, for example, if the change in redox-state affects its conformational state which in turn allows for binding to other proteins and alters their activity [63]. *Pea* mitochondrial PRX-IIF adopts a hexameric conformation in addition to its dimeric form and tightly binds thioredoxin-o [64].

### 4.3. Besides Hyperoxidation, Cys121 of PRX-IIE Is Subject to Multiple Posttranslational Modifications

Apart from oxidation and hyperoxidation, PRX-IIE is also S-glutathionylated (Figure 5 and Figure 6). Although glutathionylation of PRX-IIE inhibited its peroxidase activity (Figure 8A), S‑glutathionylation of Cys_p_ (Figure 7) may prevent PRX-IIE from hyperoxidation, as already shown for glyceraldehyde-3-phosphate-dehydrogenase (GAPDH) from spinach and isocitrate lyase from *C. reinhardtii* [65,66]. Reversal of this type of regulation of PRX‑IIE could be achieved by deglutathionylation via GRXs, TRXs, and SRXs [67,68,69]. However, the presence of GRX-S12, SRX, or GRX-C5 failed to increase the deglutathionylation rate in comparison to GSH alone, indicating that the GSH/GSSG ratio could be the main route for the regulation of this PTM in vivo.

Under normal physiological relevant concentrations of ~1 mM glutathione [70] and a ratio of 0.002% oxidized glutathione [71], it seems unlikely that PRX-IIE is glutathionylated. Application of stresses to the plant, like exposure to methylviologen or arsenic treatment, however, can alter the GSSG ratio [72,73] and, therefore, may result in S-glutathionylation of PRX-IIE. This is consistent with the results shown in Figure 9, where the application of severe stress resulted in S-glutathionylation of PRX-IIE. This post-translational modification could prevent PRX-IIE from hyperoxidation. In addition, reversible S-glutathionylation of PRX-IIE could take part in PRX-dependent signal transduction and regulation of the redox homeostasis [74].

### 4.4. PRX-IIE Binds to Target Proteins

Besides post-translational control of activity, protein-protein interactions alter the functions and properties of binding partners. PRX-IIE protein interactions mostly seem to be redox-regulated, since most of the trapped proteins were found in the fraction eluted with DTT (Figure 10A,B). In total 54 proteins were identified to interact with PRX-IIE. Since PRX-IIE is located in plastids, we focused on the 24 proteins with plastidial localization. However, the other proteins also deserve attention, since they might interact with the cytosolic type II PRXs which display high similarity with PRX-IIE (Figure 1A). Most of the identified proteins are known targets of redox regulation [75] like cyclophilin 20-3 [69] or β-carbonic anhydrase [76].

### 4.5. 14-3-3 Proteins as Binding Partner of PRX-IIE Open up New Perspectives

14-3-3 proteins function as molecular adapters and their identification as binding partners of PRX-IIE appeared interesting and novel. They are present in various isoforms in plant genomes and act as homo- and heterodimers [77]. 14‑3‑3 proteins can integrate and control multiple pathways like the abscisic acid-dependent transcription of embryo-specific target genes [78]. They participate in the regulation of salt stress tolerance and apoptotic signaling transduction [52,53]. Furthermore, they function in the development of cotyledons [54]. The 14-3-3 υ isoform co-controls the cell proliferation cycle and induces the division of chloroplasts, which results in an increased plastid number, chlorophyll content, and photosynthetic activity [54].

In this study, six 14-3-3 proteins could be identified to interact with PRX-IIE WT (Appendix A). 14-3-3 proteins are known to preferentially bind to phosphorylated motifs containing phosphoserine residues [79,80]. In addition, pThr-dependent binding as well as non-phosphorylation dependent interactions with target proteins were reported [81]. Phospho-mimetic variants of PRXIIE have been used to further address the binding properties to 14-3-3 υ under defined redox-conditions. However, preferential binding of 14-3-3 υ to these variants could not be detected under reducing as well as oxidizing conditions. Instead, the highest binding could be observed for WT PRX-IIE under oxidizing conditions (Figure 11B). In addition, introducing negative charges at positions S82, T108, and T223, resulted in an inhibition of the thiol peroxidase activity (Figure 11A). Similar results have been reported for human PRX2, where Cdk5-derived phosphorylation at T89 had a negative effect on its activity [82].

To check if the interaction of 14-3-3 υ and PRX-IIE under oxidizing conditions may alter the peroxidase activity, H_2_O_2_ reduction by PRX-IIE was determined in the presence or absence of 14-3-3 υ (Figure 11C). A tenfold excess of 14-3-3 υ increased the peroxidase activity of PRX-IIE significantly. Since such a high amount of 14-3-3 υ is necessary to alter the PRX‑IIE activity, it is more likely that the interaction is important during signaling processes. In addition, future studies like proximity-based labeling approaches, bimolecular complementation assays (BiFC), or Förster resonance energy transfer (FRET) measurements under stressed and unstressed conditions would allow for further characterization of the nature of the interaction between PRX-IIE and 14-3-3 proteins.

### 4.6. Hypothetical Outlook and Where to Go

Several mechanistic scenarios may be hypothesized. Formation of regulatory assemblies of PRX-IIE with homo- or heterodimers of 14-3-3 proteins may recruit additional binding partners. In such a regulatory complex, proximity-based oxidation between oxidized PRX-IIE and reduced 14-3-3 υ or binding of a third partner could be the regulatory mechanism that leads to changes in function and regulation of cellular processes (Figure 12).

As described previously, oxidative stress not just induces new interactions of 14-3-3 proteins with protein partners, but also results in a loss of homeostatic interactions [83]. Under oxidative stress in humans, the selenoprotein W binds to 14-3-3 with an intermolecular disulfide bridge. This process results in a release of apoptosis signal-regulating kinase-1 (ASK1) from the 14-3-3-ASK1 complex. ASK1 then activates the Jun *n*-terminal kinase and p38 MAP kinase pathways, which in turn activates caspases and thereby apoptosis [83,84]. Therefore, PRX-IIE may affect several functions of 14-3-3 υ. As indicated in the hypothetical schematics presented in Figure 12, PRX-IIE could induce dimerization of 14-3-3 υ and, thereby, mediate the binding or release of other proteins. Further studies are needed to address the importance of PRX-IIE on 14-3-3 complex formation and their associated signaling pathways. It will also be important to scrutinize the interaction of 14-3-3 proteins with the cytosolic PRXs in order to evaluate the conservation of this interaction in other cellular compartments in plants.

## 5. Conclusions

The peroxidase activity of PRX-IIE is an important route for maintaining a proper ROS homeostasis within plant cells. Our data show that the PRX-IIE activity is regulated by different redox-dependent posttranslational modifications like sulfenylation, sulfinylation, and sulfonylation, S-glutathionylation, and S-nitrosation. In addition, the interaction of PRX-IIE with members of the 14-3-3 protein family suggests a novel function of PRX-IIE in cell signaling. The interaction between PRX-IIE and other identified chloroplast proteins is highly promising and awaits further validation and scrutiny of cell biological significance.

## Figures and Tables

**Figure 1 antioxidants-10-00152-f001:**
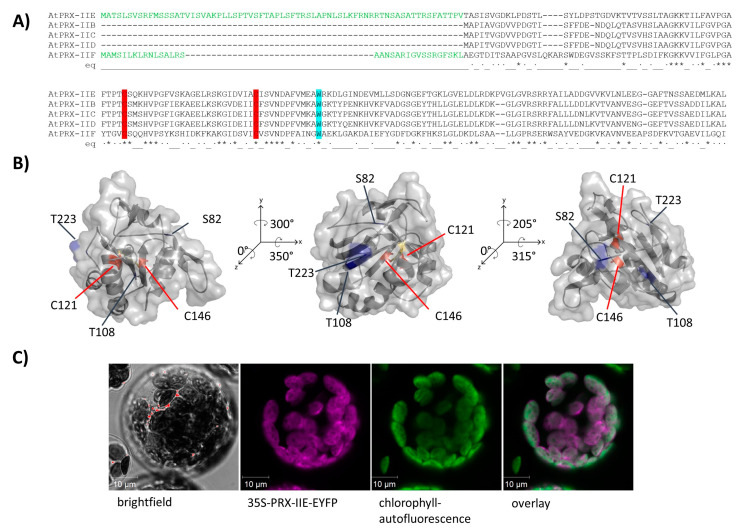
Subcellular localization of PRX-IIE. (**A**) Amino acid sequence alignment of *A. thaliana* type II PRXs. The signal peptides of PRX-IIE and PRX-IIF are highlighted in green as well as the highly conserved cysteines (C: red) and tryptophan (W: blue). (**B**) Predicted 3D structure of mature PRX-IIE from three different points of view. The model is based on the PRX D type II from *P. tremula* (pdb:1tp9A) and further analysis was done using PyMOL 2.4. Both cysteines, C121 and C146 are marked in red, whereas the putative phosphorylation sites S82, T108, and T223 are marked in blue. The distance between the cysteines is 7.83 Å. The accessible surface area of the peroxidatic cysteine at position 121 is 0.795 Å^2^. (**C**) The coding sequence of PRX-IIE was fused in-frame to EYFP as a reporter and used for transient expression in *A. thaliana* mesophyll protoplasts. Confocal laser scanning images revealed chlorophyll autofluorescence (green) and fluorescence of the PRX-IIE-EYFP construct (purple). White areas in the overlay indicate chloroplastic localization of PRX-IIE.

**Figure 2 antioxidants-10-00152-f002:**
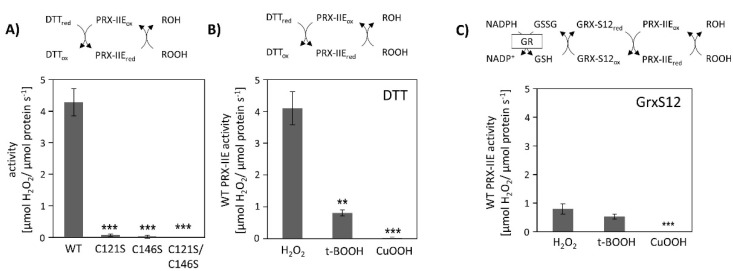
Peroxidase activity. (**A**) Peroxide reduction by wild-type PRX-IIE and its cysteine variants was measured with 400 µM H_2_O_2_, 4 mM DTT, and 2 µM protein. The decrease in H_2_O_2_ was quantified using the FOX assay. Data are means ± SD, *n* = 30 with protein of three independent protein purifications. ***: *p* < 0.001. (**B**) Substrate specificity was tested with the FOX assay with 4 mM DTT as reductant and different peroxides (400 µM H_2_O_2_, 200 µM t-BOOH, or 200 µM CuOOH). Data are means ± SD, *n* = 18–28 with protein from three independent protein purifications. **: *p* < 0.01; ***: *p* < 0.001. (**C**) PRX-IIE activity in the presence of GRX-S12 as monitored by NADPH oxidation at 340 nm in an enzyme-coupled assay. Data are means ± SD, *n* = 9 with protein from three independent protein purifications. **: *p* < 0.01; ***: *p* < 0.001.

**Figure 3 antioxidants-10-00152-f003:**
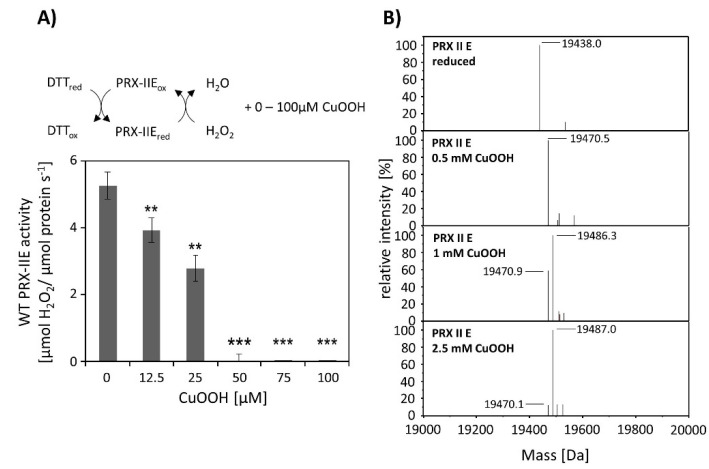
Hyperoxidation of PRX-IIE. (**A**) Activity of 2 µM PRX-IIE was determined with 400 µM H_2_O_2_ as substrate and 4 mM DTT using the FOX assay. CuOOH concentrations > 50 µM completely inhibited the peroxidase activity. Data are means of *n* = 9 ± SD with protein of three independent protein purifications. **: *p* < 0.01; ***: *p* < 0.001. (**B**) Deconvoluted ESI-MS spectra of wild-type PRX-IIE after treatment with CuOOH as described in Materials and Methods. 19,438.0 is the expected mass of the reduced (-SH), His-tagged PRX-IIE protein. The peak at 19,470.5 Da after incubation with 0.5 mM CuOOH shows a mass increase by 32 Da, which corresponds to the formation of the sulfinic acid derivative (-SO_2_H). Higher CuOOH concentrations lead to further oxidation to the sulfonic acid (-SO_3_H). Data are representative spectra of *n* = 15 measurements with the protein of three independent protein purifications.

**Figure 4 antioxidants-10-00152-f004:**
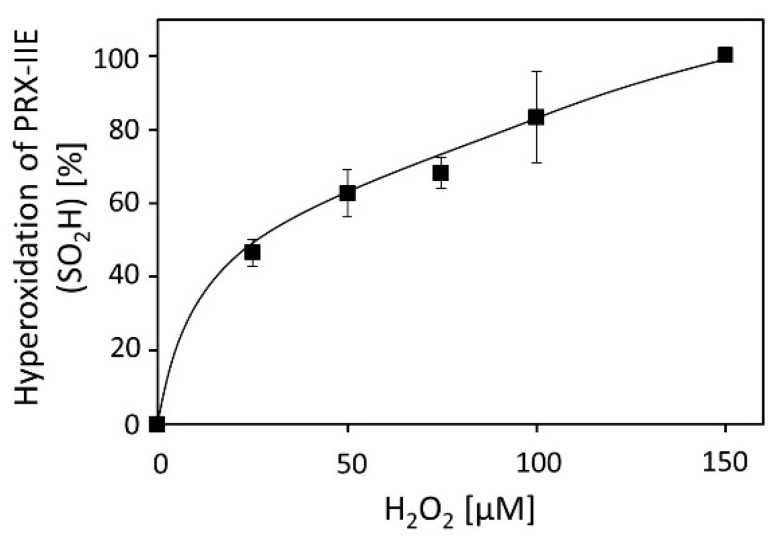
H_2_O_2_-dependent hyperoxidation of PRX-IIE. Pre-reduced and desalted PRX-IIE at 10 µM concentration was incubated with H_2_O_2_ as indicated in the presence of 500 µM DTT for 15 min. This allowed for continued re-reduction and H_2_O_2_-dependent oxidation and accumulation of the hyperoxidized form. The extent of sulfinic acid formation in percent of the total was estimated from the ratio of the peak intensities for the reduced (SH; 19.438 Da) and hyperoxidized (SO_2_H; 19.470 Da) protein in the deconvoluted ESI-MS spectra. Data are means of *n* = 10 ± SD with recombinant protein from two independent purifications.

**Figure 5 antioxidants-10-00152-f005:**
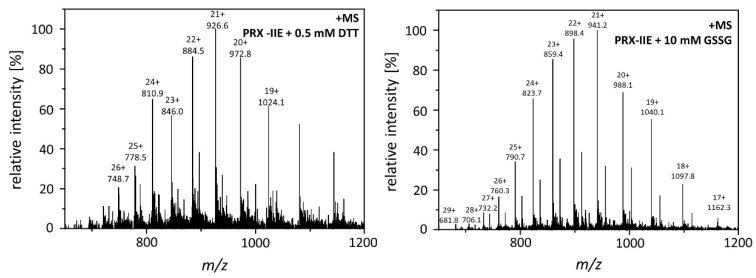
Mass spectra of reduced and S-glutathionylated PRX-IIE. S-glutathionylation was carried out by disulfide exchange with GSSG. The reaction mixtures containing 30 µM reduced PRX-IIE without (left) or with 10 mM GSSG (right) in Tris buffer, pH 8.0, were incubated at 4 °C for 18 h. The protein products were precipitated and subsequently subjected to ESI-MS analysis.

**Figure 6 antioxidants-10-00152-f006:**
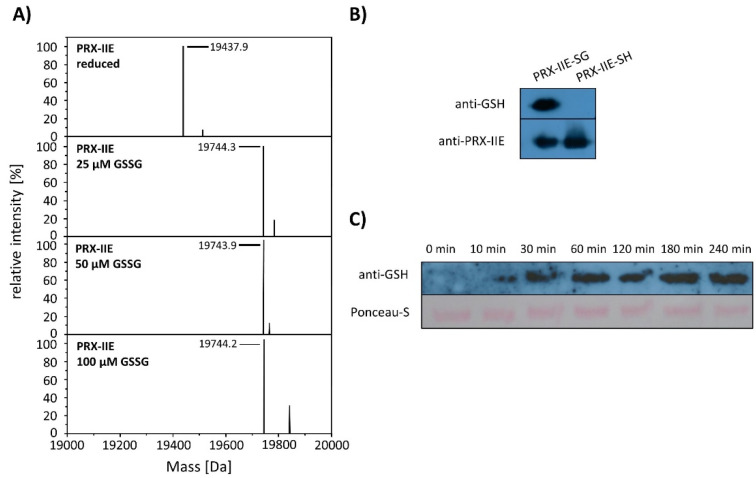
S-glutathionylation of PRX-IIE. (**A**) Deconvoluted mass spectra of reduced and S-glutathionylated PRX-IIE after treatment with oxidized glutathione (GSSG). Reduced PRX-IIE (19,438 Da) was treated with indicated GSSG concentrations and the degree of modification was estimated from the mass shift of 306 Da to 19,744 Da, which could be assigned to monoglutathionylated protein. Data are representative spectra of *n* = 12, with protein from three independent protein purifications. (**B**) Western blot analyses of reduced and S-glutathionylated PRX-IIE (treatment with 10 mM GSSG) using specific anti-GSH and anti-PRX-IIE antibody. (**C**) Time dependence of S-glutathionylation with recombinant PRX-IIE in vitro.

**Figure 7 antioxidants-10-00152-f007:**
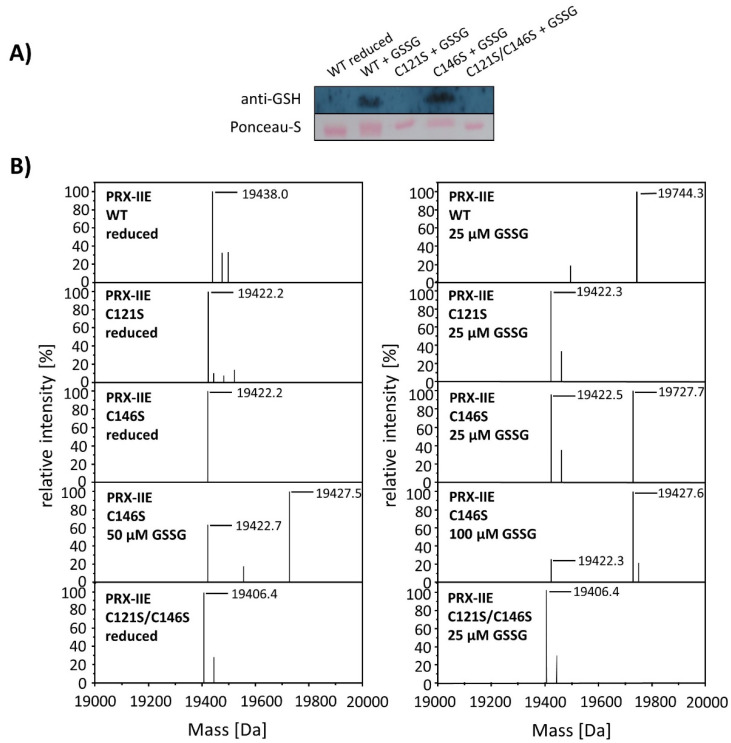
Identification of the S-glutathionylated Cys residue in PRX-IIE using Cys→Ser variants. To test for the site of glutathionylation, Cys→Ser mutated PRX-IIE variants were analyzed by Western blot with specific anti-GSH antibody and by mass spectroscopy. (**A**) 10 µM reduced WT PRX-IIE and variants (C121S; C146S; C121S/C146S) were incubated with 10 mM GSSG for 1 h at RT and subjected for Western blot analysis. (**B**) Representative deconvoluted ESI-MS spectra of PRX-IIE and variants treated with GSSG. 19438 Da corresponds to the theoretical mass of the reduced and His-tagged protein, whereas a shift of 306 Da to 19,744 was observed for the glutathionylated version of PRX-IIE with a single bound glutathione molecule. 19,422 Da corresponds to the reduced variants with single mutated cysteinyl residues (C121S; C146S) and 19,406 Da to the double mutant. Only the C146S protein, lacking the Cys_R_, was glutathionylated in a concentration-dependent manner. The figure shows representative spectra of *n* = 6 determinations, using protein from two independent purifications.

**Figure 8 antioxidants-10-00152-f008:**
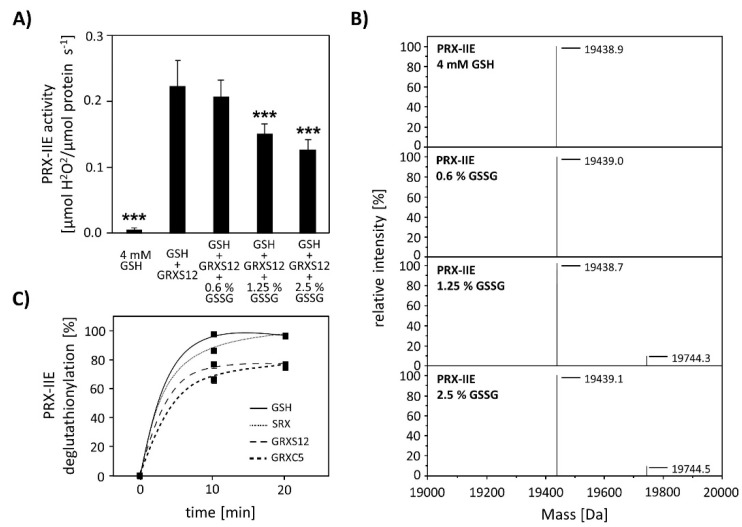
Effect of glutathionylation on peroxidase activity and deglutathionylation reaction. (**A**) Titration of peroxidase activity of PRX-IIE in the presence of GRX-S12, GSH, and H_2_O_2_ and increasing amounts of GSSG using the FOX assay. Glutathionylation and inhibition of activity occurred at <2.5% GSSG of total glutathione. Data are means of *n* = 10 ± SD with the protein of two independent protein purifications; *** *p* ≤ 0.001. (**B**) PRX-IIE was treated with different GSH/GSSG ratios and glutathionylation was determined via ESI-MS. Deconvoluted spectra reveal the presence of glutathionylated PRX-IIE already at 1.25% GSSG. (**C**) Time course of the deglutathionylation reaction. 10 µM PRX-IIE-SG were incubated together with 0.5 mM GSH and equal amounts of sulfiredoxin (SRX), GRX-S12, or GRX-C5 in 100 mM Tris-HCl, pH 8.0, at 25 °C for indicated time and was monitored using an anti-glutathione antibody. The decrease in glutathionylated PRX-IIE signal intensity relative to the 0 min time point was analyzed using ImageJ.

**Figure 9 antioxidants-10-00152-f009:**
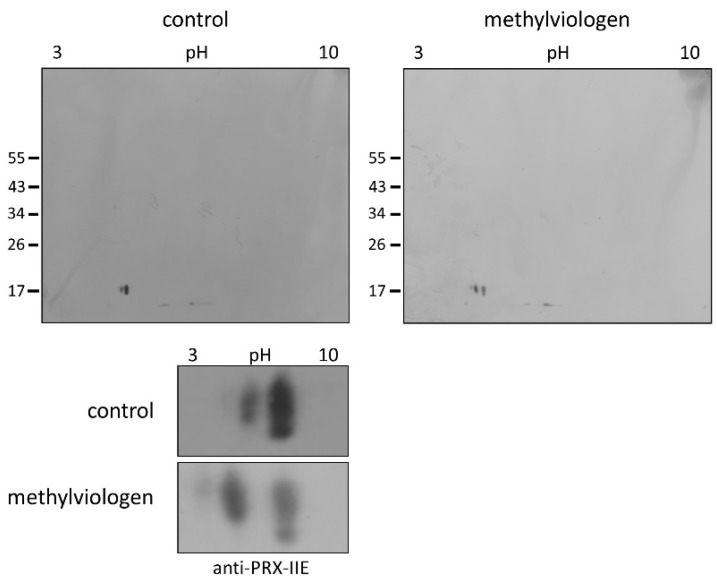
Detection of glutathionylated PRX-IIE species in vivo. *A. thaliana* plants were stressed for 3 h with 300 µM MV (blot on the right-hand side) or 0.1% (*v*/*v*) Tween-20 as control (blot on the left-hand side). S-glutathionylation was detected with specific anti-PRX-IIE antibody following separation by non-reducing 2D-SDS-PAGE and blotting. The spots of the two blots were cropped and zoomed in to clearly show the shift of the spots induced by MV treatment (the bottom part of the figure) The figure shows representative Western blots from two independent experiments.

**Figure 10 antioxidants-10-00152-f010:**
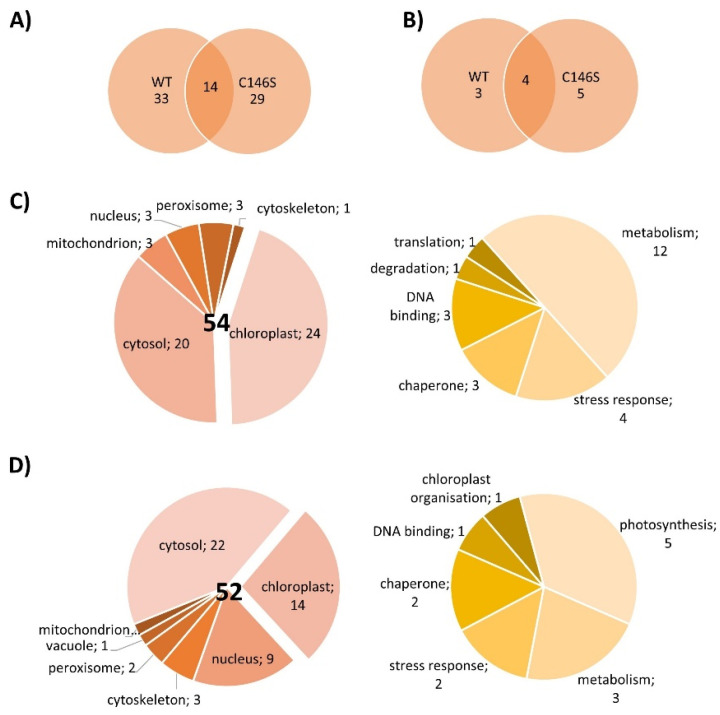
Localization and function of PRX-IIE interaction partners. Venn diagrams depicting unique and overlapping interactors of PRX-IIE WT and C146S variant after (**A**) elution with DTT and (**B**) second elution with imidazole. Identified interacting proteins with (**C**) PRX-IIE WT or (**D**) PRX-IIE C146S during first and second elution, were grouped according to their localization and function. The pie charts on the right hand side in (**C**) and (**D**) reveal the functional assignments of the chloroplast-located proteins. See also supplementary file list of identified proteins.

**Figure 11 antioxidants-10-00152-f011:**
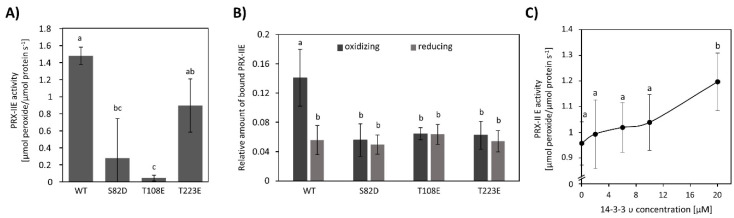
Influence of phospho-mimicry point-mutations on the activity of PRX-IIE and the interaction with 14-3-3 υ. (**A**) Peroxide reduction of WT PRX-IIE and its phospho-mimicry variants were measured with 400 µM H_2_O_2_, 4 mM DTT, and 2 µM protein. The decrease in H_2_O_2_ was quantified using the FOX assay. Data are means of *n* = 10–16 ± SD with protein from three independent protein purifications. Different letters indicate groups of significant differences (*p* ≤ 0.05) calculated by one-way ANOVA and post hoc Tukey HSD. (**B**) Overlay assay of 14-3-3 υ with PRX-IIE. A higher affinity of PRX-IIE to 14-3-3 υ was observed under oxidizing conditions. Data are means of *n* = 13–22 ± SD. Significant differences were calculated using one-way ANOVA with post hoc Tukey HSD and are indicated by different letters, whereas *p* ≤ 0.05. (**C**) Influence of 14-3-3 υ on the peroxidase activity of PRX-IIE. The activity of PRX‑IIE was measured in the presence of different concentrations of 14-3-3 υ with FOX Assay. Data are means of *n* = 12–18 ± SD. Letter a and b indicate groups of significant differences of *p* ≤ 0.05 calculated by one-way ANOVA with post hoc Tukey HSD.

**Figure 12 antioxidants-10-00152-f012:**
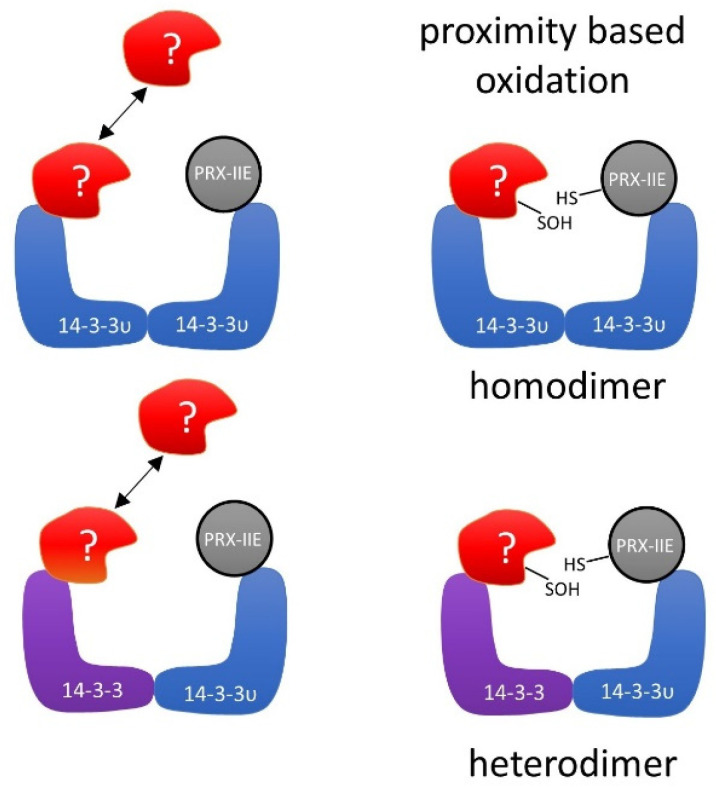
Schematic depiction of the hypothetical interaction between PRX-IIE and 14-3-3 υ. Binding of oxidized PRX-IIE to 14-3-3 υ could induce the association of a third partner. The formation of such a complex could facilitate redox regulation, e.g., by proximity-based oxidation. The hypothetical assembly may involve homo- or heterodimers of 14-3-3 isoforms.

**Table 1 antioxidants-10-00152-t001:** CuOOH-dependent thiol modifications of PRX-IIE variants. Proteins were treated with DTT or CuOOH for 1 h at RT and then analyzed by ESI-MS. Generation of the oxidized (-SOH) and hyperoxidized (-SO_2_H, -SO_3_H) forms increased the mass of the protein by 16, 32, or 48 Da. Data are means ± SD, *n* = 10 with protein from two independent purifications.

Treatment	PRX-IIE C121S		PRX-IIE C146S	
	Mass [Da]	Cys Modification	Mass [Da]	Cys Modification
**5 mM DTT** (control)	19,422.93 ± 0.61	None (SH)	19,422.22 ± 0.52	None (SH)
**0.5 mM CuOOH**	19,422.75 ± 0.57	None (SH)	19,422.57 ± 0.55 19,454.61 ± 0.44	None (SH) Hyperoxidation (SO_2_H)
**1 mM CuOOH**	19,423.13 ± 0.22	None (SH)	19,422.86 ± 0.87 19,454.93 ± 0.62	None (SH) Hyperoxidation (SO_2_H)
**2.5 mM CuOOH**	19,423.31 ± 0.17	None (SH)	19,422.46 ± 0.86 19,454.51 ± 0.92 19,471.37 ± 0.81	None (SH) Hyperoxidation (SO_2_H) Hyperoxidation (SO_3_H)
**5 mM CuOOH**	19,422.95 ± 0.54	None (SH)	19,422.52 ± 0.41 19,453.86 ± 0.53 19,470.75 ± 0.96	None (SH) Hyperoxidation (SO_2_H) Hyperoxidation (SO_3_H)
**10 mM CuOOH**	19,422.97 ± 0.57 19,438.11 ± 0.92	None (SH) Oxidation (SOH)	19,423.25 ± 0.69 19,454.14 ± 0.98 19,470.73 ± 0.81	None (SH) Hyperoxidation (SO_2_H) Hyperoxidation (SO_3_H)

**Table 2 antioxidants-10-00152-t002:** Chloroplast localized interaction partners of PRX-IIE WT. Listed are the proteins identified from elution with DTT and imidazole (grey background), with AGI code and uniport accession number.

AGI Code	Protein Accession	Protein Name
AT4G09000	P42643	14-3-3 χ
AT5G10450	P48349	14-3-3 λ
AT3G02520	Q96300	14-3-3 ν
AT5G16050	P42645	14-3-3 υ
AT3G60880	Q9LZX6	4-Hydroxy-tetrahydrodipicolinate synthase 1
AT1G02560	Q9S834	ATP-dependent Clp protease proteolytic subunit 5
AT5G03690	F4KGQ0	Fructose-bisphosphate aldolase 4
AT4G26530	O65581	Fructose-bisphosphate aldolase 5
AT5G49910	Q9LTX9	Heat shock 70 kDa protein 7
AT2G24200	P30184	Leucine aminopeptidase 1
AT5G45930	Q5XF33	Magnesium-chelatase subunit ChII-2
AT1G70890	Q9SSK5	MLP-like protein 43
AT5G26000	P37702	Myrosinase 1
AT3G62030	P34791	Peptidyl-prolyl cis-trans isomerase CYP20-3
AT2G29630	O82392	Phosphomethylpyrimidine synthase
AT5G52920	Q9FLW9	Plastidial pyruvate kinase 2
AT1G32440	Q93Z53	Plastidial pyruvate kinase 3
AT5G52520	Q9FYR6	Proline tRNA ligase
AT2G21170	Q9SKP6	Triosephosphate isomerase
AT4G17090	O23553	β-amylase 3
AT3G01500	P27140	β-carbonic anhydrase 1
AT5G14740	P42737	β-carbonic anhydrase 2
AT5G64460	Q9FGF0	Phosphoglycerate mutase-like protein 1

## Data Availability

Raw Data of the MS measurements are available at the eDAL System (http://dx.doi.org/10.5447/ipk/2021/0) as mentioned in the materials and method section of the main text.

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
