# Peer review of "Function and Regulation of Chloroplast Peroxiredoxin IIE"

_antioxidants, 2021, doi:10.3390/antiox10020152_

Round 1

Reviewer 1 Report

The authors have presented a study that investigates the activity of peroxiredoxin II in plants (A. thaliana). The study was presented in an organized fashion and the experiments were reasonably thorough and comprehensive.  The writing was a strong point. I note that this manuscript appears currently on Biorxiv.

I have the following comments:

A brief introduction to 14-3-3 nu is suggested. It just pops up for the reader in the last paragraph of the introduction.

In spite of the fact that the writing was clear, an unfortunate glitch was found throughout the manuscript. There are numerous instances where "Error! Reference source not found." appears.  This needs to be corrected.

Fig 2B. The ESI for analyzing sulfenic acid was observed only at 1 mM CuOOH. However, Fig 2A indicates that the activity was abolished at 50 uM CuOOH. Can the authors explain this observation?

Fig 7- try to improve resolution or size.  It is hard to read.

Fig 8A. The x-axis has two instances of "GSH+GRXS12+1.25% GSSG"

The spot in the control appears darker in the entire 2-D plot compared to the MV-treated 2D WB.  The zoomed figures intensities do not appear the same...could the authors explain this?

Minor:

p.4,l173 - chromatographie -> chromatography

p6, l229. are -> is

p7,l261 - depende^n^t

Author Response

The authors have presented a study that investigates the activity of peroxiredoxin II in plants (A. thaliana). The study was presented in an organized fashion and the experiments were reasonably thorough and comprehensive.  The writing was a strong point. I note that this manuscript appears currently on Biorxiv.

 I have the following comments:

  1. A brief introduction to 14-3-3 nu is suggested. It just pops up for the reader in the last paragraph of the introduction.

Thanks to the reviewer, this is a good point, we addressed this issue by adding some information about the 14-3-3 protein in lines 81-85.

  1. In spite of the fact that the writing was clear, an unfortunate glitch was found throughout the manuscript. There are numerous instances where "Error! Reference source not found." appears.  This needs to be corrected.

We are sorry you experienced this problem with our manuscript. Since we do not find these glitches within our word document or PDF file, we assume this happened because of some word settings. To address this, we disabled the automatic actualisation of fields in our word document and hope this solves the problem.

  1. Fig 2B. The ESI for analyzing sulfenic acid was observed only at 1 mM CuOOH. However, Fig 2A indicates that the activity was abolished at 50 uM CuOOH. Can the authors explain this observation?

Thanks to the reviewer. This is a good point. Although incubation of PRX-IIE with 0.5 mM CuOOH (Figure 3B) results in the formation of the sulfenic acid, it is still a higher concentration of CuOOH, than used in the peroxidase activity measurement in Figure 3A. Both assays differ: Figure 3A shows an activity measurement in the presence of DTT, H2O2 and CuOOH, therefore, PRX-IIE continuously undergoes the catalytical cycle and slowly accumulates the sulfinic acid derivative as described by Koenig et al (2003). In a converse manner, PRX-IIE in Figure 3B was only incubated with CuOOH. Hyperoxidation requires two reactions of the CysP with the peroxide, namely once to form the sulfenic acid derivative, subsequently the sulfenic acid must react with another peroxide before the resolving Cys can form the disulfide bridge. This ‘double hit’-reaction efficiently occurs only at very high concentrations. We address this point and added it accordingly in lines 515-518

  1. Fig 7- try to improve resolution or size.  It is hard to read.

Done

  1. Fig 8A. The x-axis has two instances of "GSH+GRXS12+1.25% GSSG"

Done

  1. The spot in the control appears darker in the entire 2-D plot compared to the MV-treated 2D WB. The zoomed figures intensities do not appear the same...could the authors explain this?

Thanks to the reviewer. We checked again the blots and found a change in contrast and brightness in the zoomed in blots and corrected it. This differences in spot intensity between the blot of the control and the MV treated plant material is due to the method. We develop Western Blots using X-ray films. Although we used the same exposure time for both blots, small difference during development of the film or even the incubation with the antibody solutions can result in different intensities of the spots.

Minor:

  1. 4,l173 - chromatographie -> chromatography

Done

  1. p6, l229. are -> is

Done

  1. p7,l261 - depende^n^t

Done

Reviewer 2 Report

Unfortunately the work is not read easily;the key points of the research are not clearly evident. Many gaps are found in the description of the method of cell fractionation and the control of the purity of the fractions, so the results obtained do not seem certain to me.

Author Response

Unfortunately the work is not read easily;the key points of the research are not clearly evident. Many gaps are found in the description of the method of cell fractionation and the control of the purity of the fractions, so the results obtained do not seem certain to me.

Thanks to the reviewer, we generally revised the whole manuscript according to the detailed suggestion made by the other four referees. Since we deleted Figure 1D we removed the description of the cell fractionation within the material and method part.

Reviewer 3 Report

The research article by Dreyer et al. reveals the function and regulation of chloroplast peroxiredoxin IIE using proteomic approach and interaction studies. The manuscript reads well, and all sections well delineated.
However, conclusion section is missing. The author should conclude their finding in the conclusion section.

Author Response

The research article by Dreyer et al. reveals the function and regulation of chloroplast peroxiredoxin IIE using proteomic approach and interaction studies. The manuscript reads well, and all sections well delineated.
However, conclusion section is missing. The author should conclude their finding in the conclusion section.

Thanks to the positive comment of the reviewer. Addition of a conclusion section is a good point, therefore we added it to the manuscript (lines 614-621)

Reviewer 4 Report

Manuscript number:  antioxidants-1008472 “Function and regulation of chloroplast peroxiredoxin IIE”

 In this manuscript, Dreyer and collaborators report on the chloroplast peroxiredoxin IIE (PRXIIE). The authors have characterized several aspects of its regulation and suggested a new function. They have used a number of techniques and experimental procedures and obtained results related to the regulation of the PRXIIE.  In addition after identifying several target proteins they selected one to try to discover new functions. According to their results, the authors show that PRXIIE is subjected to multiple posttranslational modifications and suggest the participation of 14-3-3 Ê‹ that interact with PRXIIE in the redox regulation of certain target proteins.

The work addresses an interesting topic shared with other research group in this moment in relation with the redox regulation in plants and especially in the chloroplasts. This work is part of the main research lines of the group.

In general, most of the experiments and analysis are well conducted but others are partially incomplete. The idea of looking at both regulation and function of the PRXIIE together is interesting but raises several problems. Among them, some of the experiments are not analysed in depth and the results obtained needs to be confirmed, and an example is to demonstrate the interaction between the PRXIIE and the 14-3-3 Ê‹ protein, perhaps with BiFC assay? The suggested role of this protein is a speculation which is understandable, that still needs to be demonstrated. The fact that half of the proteins that have interacted with PRX are localized in the chloroplast suggests that they may not all be specific. I wonder why the authors have not analysed the proteins that interacted with the C146S variant? Could be more specific? Perhaps the identification of new PRXIIE protein targets and the function could have been completed and presented separately.

In my opinion, despite this manuscript is interesting and provides a number of valuable data that were organized nicely, it looks like it is between a review (first result chapter) and a multiple research article, which sometimes generates a bit of blur on the different objectives, PRXII activity, posttranslational modifications and identification of new target proteins to discover specific functions. Perhaps it would have been more useful to dedicate the study in more depth to only one aspect of the proposals.

The text is well written, however I have concerns about some parts of the manuscript, such as that few paragraphs of results need more description and details, some of the figures needs to be completed or improved; I provide some general and specific comments following the order of the edition.

 1- Page 5, line 209, “Besides the mitochondrial PRX-IIF, RRX-IIE is…” should be corrected.

 2- Page 6, Figure 1A is difficult to read (even on the screen). Figure 1D western blot would be clearer if the protein control PRX IIE is included. The localization in the chloroplast was already known and the first part of the study would not have been necessary.

 3- Page 7, line 264 (Figure 2B) should be (Figure 3B)?

 4- Page 11, lines 359-371. The western blot could be shown? And explained better in the text.

5- page 12, Figure 8A 1.25% is repeated twice,  last column should be 2.5%?

 6- Page 13, Figure 9, could the authors describe the different figures in the legend.

  7- Page 13, 3.5 paragraph; identified specific targets in isolated chloroplasts would have been an alternative to avoid contamination from other organelles and part of the cell?

 8- Page 14 lines 430-432; explain why 14-3-3 Ê‹ was chosen as a representative. Availability? Does the protein was already purified and from which plant?

 9- Page 16, lines 493-494, page 17, lines 513-515 and paragraph 4.6. Until it can be demonstrated these proposals are only suggestions and hypothesis.  

Author Response

Reviewer 4

Manuscript number:  antioxidants-1008472 “Function and regulation of chloroplast peroxiredoxin IIE”

 In this manuscript, Dreyer and collaborators report on the chloroplast peroxiredoxin IIE (PRXIIE). The authors have characterized several aspects of its regulation and suggested a new function. They have used a number of techniques and experimental procedures and obtained results related to the regulation of the PRXIIE.  In addition after identifying several target proteins they selected one to try to discover new functions. According to their results, the authors show that PRXIIE is subjected to multiple posttranslational modifications and suggest the participation of 14-3-3 Ê‹ that interact with PRXIIE in the redox regulation of certain target proteins.

The work addresses an interesting topic shared with other research group in this moment in relation with the redox regulation in plants and especially in the chloroplasts. This work is part of the main research lines of the group.

  1. In general, most of the experiments and analysis are well conducted but others are partially incomplete. The idea of looking at both regulation and function of the PRXIIE together is interesting but raises several problems. Among them, some of the experiments are not analysed in depth and the results obtained needs to be confirmed, and an example is to demonstrate the interaction between the PRXIIE and the 14-3-3 Ê‹ protein, perhaps with BiFC assay? The suggested role of this protein is a speculation which is understandable, that still needs to be demonstrated. The fact that half of the proteins that have interacted with PRX are localized in the chloroplast suggests that they may not all be specific. I wonder why the authors have not analysed the proteins that interacted with the C146S variant? Could be more specific? Perhaps the identification of new PRXIIE protein targets and the function could have been completed and presented separately.

Thanks to the reviewer. BiFC and FRET measurements or even proximity based labelling approaches are a great idea to perform in future studies. We included this idea in the manuscript in lines 587-591.

It is a good point to address also the proteins that interacted with the C146S variant, but in this study, we were mainly interested whether PRX-IIE plays a role in cell signalling through direct protein-protein interactions, therefore we focused on proteins interacting with PRX-IIE WT, based on electrostatic interaction. Proteins that covalently bind PRX-IIE should not be trapped by the WT but by the C146S variant, therefore we did not choose a protein trapped by the C146S variant.

  1. In my opinion, despite this manuscript is interesting and provides a number of valuable data that were organized nicely, it looks like it is between a review (first result chapter) and a multiple research article, which sometimes generates a bit of blur on the different objectives, PRXII activity, posttranslational modifications and identification of new target proteins to discover specific functions. Perhaps it would have been more useful to dedicate the study in more depth to only one aspect of the proposals.

We addressed the issue that the first results chapter seems like a review by shortening this chapter (lines 225-226; 231-233). We fully understand the thinking of this expert referee. It appears to us that readers and editors nowadays wish to see studies addressing several aspects of a process or mechanism. This original paper addresses entirely new features of PRX-IIE. In order to provide a rather complete view, we prefer to keep the structure of the paper.

  1. The text is well written, however I have concerns about some parts of the manuscript, such as that few paragraphs of results need more description and details, some of the figures needs to be completed or improved; I provide some general and specific comments following the order of the edition.
    • Page 5, line 209, “Besides the mitochondrial PRX-IIF, RRX-IIE is…” should be corrected.

Done

  • Page 6, Figure 1A is difficult to read (even on the screen). Figure 1D western blot would be clearer if the protein control PRX IIE is included. The localization in the chloroplast was already known and the first part of the study would not have been necessary.

Thanks for the comment, since localization is already known we removed Figure 1D and also shortened this result section to compromise the information.

  • Page 7, line 264 (Figure 2B) should be (Figure 3B)?

Done

  • Page 11, lines 359-371. The western blot could be shown? And explained better in the text.

We included the Figure in the Supplement material (Figure S1). Furthermore, we added some explanations in lines 389-391 and in die Figure 8 caption lines 403-405.

  • page 12, Figure 8A 1.25% is repeated twice,  last column should be 2.5%?

Done

  • Page 13, Figure 9, could the authors describe the different figures in the legend.

We added appropriate information to the Figure 9 legend. Thanks for the comment.

  • Page 13, 3.5 paragraph; identified specific targets in isolated chloroplasts would have been an alternative to avoid contamination from other organelles and part of the cell?

Good idea, thank you for the comment. It makes sense to use isolated chloroplasts to identify target proteins, but as mentioned in lines 559 & 560 the other proteins, localized in the cytosol could be also interesting interaction partners in case of the cytosolic type II Peroxiredoxins (PRX-II B, C and D) due to the high similarity with PRX-IIE.

  • Page 14 lines 430-432; explain why 14-3-3 Ê‹ was chosen as a representative. Availability? Does the protein was already purified and from which plant?

We choose 14-3-3 υ as representative because of the high number of identified unique peptides and the chloroplast localisation reported by Sehnke et al. (2000). We added this information in lines 455-458.

  • Page 16, lines 493-494, page 17, lines 513-515 and paragraph 4.6. Until it can be demonstrated these proposals are only suggestions and hypothesis.  

We clarified in the manuscript, that these issues are suggestions and hypothesis. Also, the first sentence of Section 4.6 mentions already that the scenarios described in the section are just hypothesis.

Reviewer 5 Report

The manuscript entitled ”Function and regulation of chloroplast peroxiredoxin IIE” is a very interesting and well-written paper which publication I endorse. I have just a few comments that can improve the manuscript.

Comments:

  1. Statistical analysis subsection is missing from Materials & Methods.
  2. Could the authors add a better blot for Figure 6 C?
  3. Why did the authors express results of FOX-assay as means ± SE in Fig. 11 A while in others are means ± SD? In addition, since the authors are comparing more than 2 groups 1-way ANOVA would be more suitable than a Student t-test.
  4. In Fig. 11 B and C, the authors state “Significant differences were calculated using Student t-test and are indicated by different letters p ≤ 0.05.” but it is not clear what the letters a, b, and c are comparing.
  5. Some typos could be found across the text such as Affinitiy chromatographie, RRX-IIE, etc.

        In addition, abbreviations should be uniform throughout the text.

Author Response

The manuscript entitled ”Function and regulation of chloroplast peroxiredoxin IIE” is a very interesting and well-written paper which publication I endorse. I have just a few comments that can improve the manuscript.

Comments:

  1. Statistical analysis subsection is missing from Materials & Methods.

Thanks to the reviewer, we included a statistical analysis chapter into the material and method section (lines 214-220).

  1. Could the authors add a better blot for Figure 6 C?

We changed the color settings of the blots shown in Figure 6 and 7. Instead of using grayscale we now included the original colored blots, which improves the visibility of the spots.

  1. Why did the authors express results of FOX-assay as means ± SE in Fig. 11 A while in others are means ± SD? In addition, since the authors are comparing more than 2 groups 1-way ANOVA would be more suitable than a Student t-test.

Thanks to the reviewer. We changed the figure 11 A and changed the error to standard deviation. Furthermore, we recalculated the significant differences using one-way ANOVA with post hoc Tukey HSD as you suggested.

  1. In Fig. 11 B and C, the authors state “Significant differences were calculated using Student t-test and are indicated by different letters p ≤ 0.05.” but it is not clear what the letters a, b, and c are comparing.

Thank you for your comment. We also recalculated the significant differences with one-way ANOVA between each data set. Therefore, we also adjusted the Figure 11 legend.

  1. Some typos could be found across the text such as Affinitiy chromatographie, RRX-IIE, etc.

Done

        In addition, abbreviations should be uniform throughout the text.

Thanks to the reviewer, we checked again the whole manuscript regarding abbreviations.

Round 2

Reviewer 2 Report

Dear authors, I see that you  followed the suggestions of the referees and that you made the necessary changes, so the work has increased its scientific potential. I agree to proceed with the acceptance of the manuscript for its publication.

Reviewer 3 Report

The authors have fully addressed reviewer’s suggestion and critique during the first rounds of peer review. This reviewer is satisfied with the changes made by authors.  I would recommend this manuscript for publication.

Reviewer 4 Report

Manuscript number:  antioxidants-1008472 “Function and regulation of chloroplast peroxiredoxin IIE”

The authors have responded to each of the questions and have modified and completed the text according to most of the suggestions. The identification of potential target proteins opens up a wide field of possibilities that must be investigated exhaustively and in detail to reveal specific functions of the PRXIIE. It is indeed probably not easy to manage several results related to regulation and function of the protein, and sometimes it is a bit difficult to follow. Nevertheless, this manuscript provides very interesting information.

I appreciate the changes made in the figures and related texts for a better and easier comprehension.
